# Rearrangement of 3D genome organization in breast cancer epithelial to mesenchymal transition and metastasis organotropism

Priyojit Das[1†‡], Rebeca San Martin[2†§], Tian Hong[1,2#], Rachel Patton McCord[1,2]*

[1]UT-ORNL Graduate School of Genome Science and Technology, University of Tennessee, Knoxville, United States; [2]Biochemistry & Cellular and Molecular Biology, University of Tennessee, Knoxville, United States

**\*For correspondence:**
rmccord@utk.edu

†These authors contributed equally to this work

**Present address:** ‡Department of Molecular Biology, Massachusetts General Hospital; Department of Genetics, Harvard Medical School, Boston, United States; §Department of Cell Biology, Department of Oncology, and Ruth L. and David S. Gottesman Institute for Stem Cell and Regenerative Medicine Research, Albert Einstein College of Medicine; Cancer Dormancy Institute, Montefiore Einstein Comprehensive Cancer Center, Bronx, United States; #Department of Biological Sciences, The University of Texas at Dallas, Richardson, United States

**Competing interest:** The authors declare that no competing interests exist.

## eLife Assessment

This **valuable** study explores the role of spatial genome organization in oncogenic transformation, addressing an ambitious and significant topic. The authors have assembled comprehensive datasets from various subtypes of localized and lung-metastatic breast cancer cells, as well as from healthy and cancerous lung cells. They identified switching patterns in the 3D genome organization of lung-metastatic breast cancer cells, revealing a reconfiguration of genome architecture that resembles that of lung cells. This provides **solid** evidence with significant biomedical implications for epigenetic regulation in both normal physiology and disease.

**Abstract** Human breast cancer cells exhibit organotropism during metastasis, showing preferential homing to certain organs such as bone, lung, liver, and brain. Spatial genome organization plays a crucial role in oncogenic transformation and progression, but the extent to which chromosome architecture contributes to organ-specific metastatic traits is unclear. This work characterizes chromosome architecture changes associated with organotropic metastatic traits. By comparing a collection of human genomic data from different subtypes of localized and lung metastatic breast cancer cells with both normal and cancerous lung cells, we find important trends of genomic reorganization. The most striking differences in 3D genome compartments segregate cell types according to their epithelial vs. mesenchymal status. This epithelial–mesenchymal transition (EMT) compartment signature occurs at genomic regions distinct from transcription-defined EMT signatures, suggesting a separate layer of regulation. Specifically querying organotropism, we find 3D genome changes consistent with adaptations needed to survive in a new microenvironment, with lung metastatic breast cancer cells exhibiting compartment switch signatures that shift the genome architecture to a lung cell-like conformation and brain metastatic prostate cancer cells showing compartment shifts toward a brain-like state. TCGA patient data reveals gene expression changes concordant with these organ-permissive compartment changes. These results suggest that genome architecture provides an additional level of cell fate specification informing organotropism and enabling survival at the metastatic site.

## Introduction

Breast cancer is the leading malignant disease in women worldwide. It is highly heterogeneous; based on molecular characteristics, it can be divided into four broad subgroups: luminal A (estrogen receptor

**eLife digest** Nearly all cancers have the potential to spread to other parts of the body. Their ability to metastasize via tissues and blood or lymph vessels depends on several factors, such as the type, size, and location of the primary tumor. But metastasis is not a random process, and tumor cells often exhibit a preference for colonizing specific organs – a phenomenon also known as organotropism. In the case of breast cancer, these preferred sites are the lungs, liver, brain, and bones.

The mechanisms by which cells survive in a new, drastically different tissue environment remain poorly understood. One theory is that the structure of chromosomes inside cells, which plays a determining role in cell function and identity, may explain how tumors adapt to different tissues.

Das et al. used a new genome-folding-centric approach to find out why breast cancer metastasizes to certain organs, such as the lung. The researchers analyzed genome compartmentalization data of various human cancer cell lines, including breast cells (both localized cancer and lung-metastatic cancer cells) and lung cells (healthy and cancerous ones).

The results showed that the three-dimensional arrangement of DNA within the cell nucleus might play a role in a tumor's metastatic behavior. One aspect of genome structure is that the way in which long DNA strands are packed and folded inside the human nucleus is not random. Chromosome regions are spatially separated into two compartments: one active, containing genes that are available to be transcribed or used, and another inactive, where genes are tightly packed and silent. As cancer cells become invasive and enter a migratory state, specific parts of the genome switch between these compartments. This spatial reorganization influences gene accessibility and impacts the preference for metastatic sites. The most significant finding is that lung-metastatic breast cancer cells acquire a genome architecture that structurally resembles features of normal lung cells. Further, these changes in compartment identity are not always accompanied by increases in gene transcription. The study by Das et al. contributes to a broader understanding of how the three-dimensional genome may contribute to metastatic transformation of cancerous cells and their ability to colonize distant sites. It also highlights that certain genome compartmentalization signatures might be used as a prognostic biomarker. However, significant work in translational and clinical research would be required before these benefits could be realized. The next critical step is to validate these genome changes and their role in organotropism in patient-derived samples. Further studies are also needed to identify the specific mechanisms that regulate and control the compartment switches.

ER+ progesterone receptor PR+/− human epidermal growth factor receptor 2 HER2−), luminal B (ER+ PR+/− HER2+), HER2-enriched (ER− PR− HER2+), and triple-negative/basal (ER− PR− HER2−) (*Fragomeni et al., 2018*). About 20–30% of early-stage breast cancer patients have a chance to ultimately develop distant metastasis, which accounts for 90% of all breast cancer-related deaths (*Chaffer and Weinberg, 2011*). Overall, metastasis implies the dissemination of cancer cells from the primary tumor site into the circulation and later extravasation to a new site which leads to the formation of a secondary tumor. However, the organs affected by breast cancer metastasis vary remarkably, depending on the disease subtype, molecular features, primary tumor location, and secondary site microenvironment. This phenomenon is known as 'metastasis organotropism' (*Chen et al., 2018*). For example, breast cancer cells can metastasize to the bone, lung, liver, and brain. However, bone metastasis accounts for a major percentage of metastatic cases compared to other organs (*Savci-Heijink et al., 2016*). Moreover, luminal A and basal subtypes of breast cancer show a higher incidence of bone and lung metastasis, respectively, whereas luminal B preferentially metastasizes to both organs (*Smid et al., 2008*). Organotropic behavior is congruent with the 'seed and soil' hypothesis, according to which a cancer cell (seed) having specific intrinsic abilities will only be able to survive in a host local microenvironment (soil) that is favorable to it (*Langley and Fidler, 2011*).

Secondary organ colonization by circulating tumor cells is a complex process, ruled by dynamic and evolving interactions between cancer cells and the host microenvironment. In most situations, the microenvironment is unfavorable, and the disseminated cells fail to survive in that 'soil'. Despite the low survival rate of cancer cells during the metastatic colonization process, cells with the intrinsic capabilities to interact with the host microenvironment can survive and proliferate in the secondary site (*Obenauf and Massagué, 2015*). Alternatively, cells might arrive at the secondary site and remain

dormant under the influence of the local microenvironment (*Sharma et al., 2016*; *Taichman et al., 2013*; *Shiozawa et al., 2016*; *Dalla et al., 2023*) slowly acquiring characteristics that would later allow their proliferation. Importantly, the microenvironment surrounding primary breast tumors seems to condition the cells to specific metastatic profiles. For example, if the primary tumor stroma is enriched in transforming growth factor beta (TGFB), lung metastasis prevails through the induction of angiopoietin-like 4 (*Padua et al., 2008*). In contrast, a tumor stroma enriched with mesenchymal cytokines like C-X-C motif chemokine 12 (CXCL12) and insulin-like growth factor 1 (IGF1) predisposes primary cancer cell population to home into the bone marrow (*Zhang et al., 2013*). How these different reactive microenvironments condition the origin of specific metastatic traits still poses an outstanding question.

The current view in the cancer biology field is that transcriptional gene signatures associated with organ-specific metastatic traits are regulated in part by cues from the reactive microenvironment that accompanies primary tumorigenesis (*Padua et al., 2008*; *Zhang et al., 2013*; *Sceneay et al., 2012*; *Gao et al., 2019*). However, whether regulating those genes involves changes in 3D genome organization is still an open question. Recent advancements in imaging and molecular biology techniques have shown that the genome organizes in a non-random multi-layered fashion inside the nucleus, and each layer of organization contributes both independently and synergistically to gene expression regulation (*McCord et al., 2020*). At a large scale, the reorganization of genome compartments, in which euchromatic and heterochromatic genomic regions are spatially segregated, is also found to be associated with cancer progression (*Achinger-Kawecka et al., 2020*; *Kim et al., 2022*; *San Martin et al., 2022*). It has been shown that as breast cancer becomes endocrine-resistant, the change in compartment organization is accompanied by gene expression changes and differential estrogen receptor binding (*Achinger-Kawecka et al., 2020*). Other studies have also found that an unintended consequence of various drug treatments and therapies is the rearrangement of chromatin architecture at different levels (*Achinger-Kawecka et al., 2020*; *Barutcu et al., 2016*; *Yang et al., 2020*). Though these types of studies are essential for elucidating the association between the reorganization of the genome in three dimensions and oncogenic transformation, they do not consider the genome organization changes experienced by the metastatic cell that occur as a result of the interaction with the microenvironment of the secondary site. Genome spatial compartmentalization is strongly associated with cell type definition: genomic regions segregated into the A (euchromatic) or B (heterochromatic) compartments differ among cell types and tissue origin (*Li et al., 2024*). These compartmentalization profiles help encode stable cell fate gene expression (*Schmitt et al., 2016*; *Vilarrasa-Blasi et al., 2021*; *Owen et al., 2022*). Therefore, we hypothesize that similar to gene expression, metastatic cancer cells experience 3D genome structure rearrangements that mimic the compartment signatures of their new host microenvironment. To test this hypothesis, here we focused on the breast cancer–lung metastasis system. We compared genome organization data from different types of breast epithelial cell lines (normal, localized cancer, and metastatic) with normal and cancerous lung cells. Our analysis revealed that metastatic breast cancer cells have a higher proportion of lung-specific spatial compartment changes than localized breast cancer cells. We find similar brain microenvironment-specific compartment changes in prostate cancer cells that have the potential to metastasize to the brain. These distal-site concordant compartment shifts often correspond to changes in gene expression in both cell lines and TCGA patient samples. However, we find that 3D genome changes do not always correlate with gene expression changes. Instead, particularly across the epithelial–mesenchymal transition (EMT) spectrum, we find a set of genes that change expression without needing spatial reorganization and then a distinct set of epithelial and mesenchymal phenotype genes that change their spatial compartment without altered gene expression. We hypothesize that spatial genome organization facilitates immediate changes in gene expression in certain loci but can also poise certain regions for future activation in a way that favors EMT and adaptation to metastatic organ microenvironments.

## Results

### Modeling the breast cancer lung metastasis system

To investigate whether breast cancer spatial genome organization showed rearrangements specific to a distant organ metastatic microenvironment, we needed to assemble genome spatial compartment data from different types of normal and cancerous cells for both breast and a secondary organ. Since

most breast cancer subtypes show a preference for lung metastasis, we first focused on this metastatic site (*Bennett et al., 2022*). We selected a cohort of breast and lung cell lines, both normal and cancerous, for which publicly available genome organization (Hi-C) is available (*Supplementary file 1*). A total of 15 cell lines with available data, including primary cells and cancer cell lines from both organs, were selected. A classification of the cells used in this study based on their malignancy status and other molecular signatures is shown in *Figure 1a*. Briefly, for the breast cancer system, we considered both HMEC and MCF10A as normal (non-tumorigenic) breast epithelial cells and placed them at the starting node of the model. Cancer cell lines were then divided into three subcategories based on breast cancer molecular signatures: luminal (ZR751, BT474, ZR7530, MCF7, and T47D), HER2-enriched (HCC1954 and SKBR3), and triple-negative (HCC70, BT549, and MDA-MB-231). We then assigned cell lines to 'localized' or 'metastatic' status based on the site from which they were originally collected in the patient (see details in *Supplementary file 1*; *Engel and Young, 1978*; *Gazdar et al., 1998*; *Wang et al., 2021*; *Stone et al., 1978*). Specifically, within the luminal subtype, ZR751 and BT474 were designated as localized cancers, and the remaining three cell lines, ZR7530, MCF7, and T47D, were used as models for ascites, lung, and lung metastatic breast cancers, respectively. Similarly, for the HER2-enriched subcategory, HCC1954 and SKBR3 were considered localized and lung metastatic cancers, respectively. Finally, in the case of the triple-negative subtype, HCC70 and BT549 were used to model localized cancer whereas MDA-MB-231 represents lung metastatic cancer. For the lung portion of the system, HTBE primary cells represent normal lung epithelium, and both the A549 and H460 cell lines represent localized lung cancer. The valuable MetMap study in mouse has measured the organs that can be colonized by human-derived cancer cell lines when they are injected into mice (*Jin et al., 2020*). The cell lines we consider here may have different or additional 'potential' metastatic sites in this type of mouse model. But, it is known that species differences can lead to divergent organotropism in mouse models as compared to human patients (*San Martin et al., 2017*; *Simons et al., 2019*). Therefore, we use the site of collection as compared to the site of origin in the human patient as our reference point for primary and secondary site organs.

## Subtype and epithelial–mesenchymal state-specific signatures are present in breast cancer compartmental organization

We compared the supervised ordering of the cell lines described above with the ordering derived from unsupervised analysis of genome-wide A/B compartment organization data. Compartment identity strengths were calculated using eigenvector decomposition on the 250 kb binned Hi-C data for each cell line (Methods). In this analysis, a positive value signifies a euchromatic A compartment region, while a negative value represents a heterochromatic B compartment association.

When we applied hierarchical clustering on the genome-wide compartment data, we observed a grouping of cells that was congruent with the cell ordering constructed based on molecular and clinical features (*Figure 1b*). For example, all the normal cell types, including both the breast and lung systems, were clustered together. Similarly, cells from the luminal subtype and the lung carcinoma cells were grouped in their respective clusters. However, we also observed a few interesting exceptions from the clustering result. MDA-MB-231, a breast cancer cell line isolated from lung pleural effusion, was grouped with the lung carcinoma cell lines, suggesting its compartment organization shares more features with lung cancer than do other breast cancer cell lines. We also found that HCC70 and HCC1954, which were both derived from localized cancer, clustered relatively close to the normal epithelial cells. Collectively, this unsupervised analysis shows that the different subtypes of cancer exhibit distinctive 3D genome organization features at the compartment level (*Figure 1b*), as previous studies suggested (*Kim et al., 2022*).

Performing principal component analysis (PCA) on the genome-wide compartment data revealed similar trends (*Figure 1c*). Interestingly, the ordering of cell types along the first principal component (PC1) captured a transition between epithelial and mesenchymal-like cells. Cells characterized as highly epithelial were found on the left extreme of the PC1 axis, and as we move toward the right along PC1, the cells acquire a mesenchymal phenotype. For example, MDA-MB-231 cells are derived from healthy cells of epithelial origin but have undergone an EMT and show a mesenchymal-like phenotype (*Charafe-Jauffret et al., 2006*; *Yin, 2011*). In our Hi-C principal components analysis, this cell line has a highly positive PC1 value. In contrast, MCF7 cells maintain a more epithelial morphology and are found at the opposite end of the PC1 spectrum from MDA-MB-231. The organization of cell

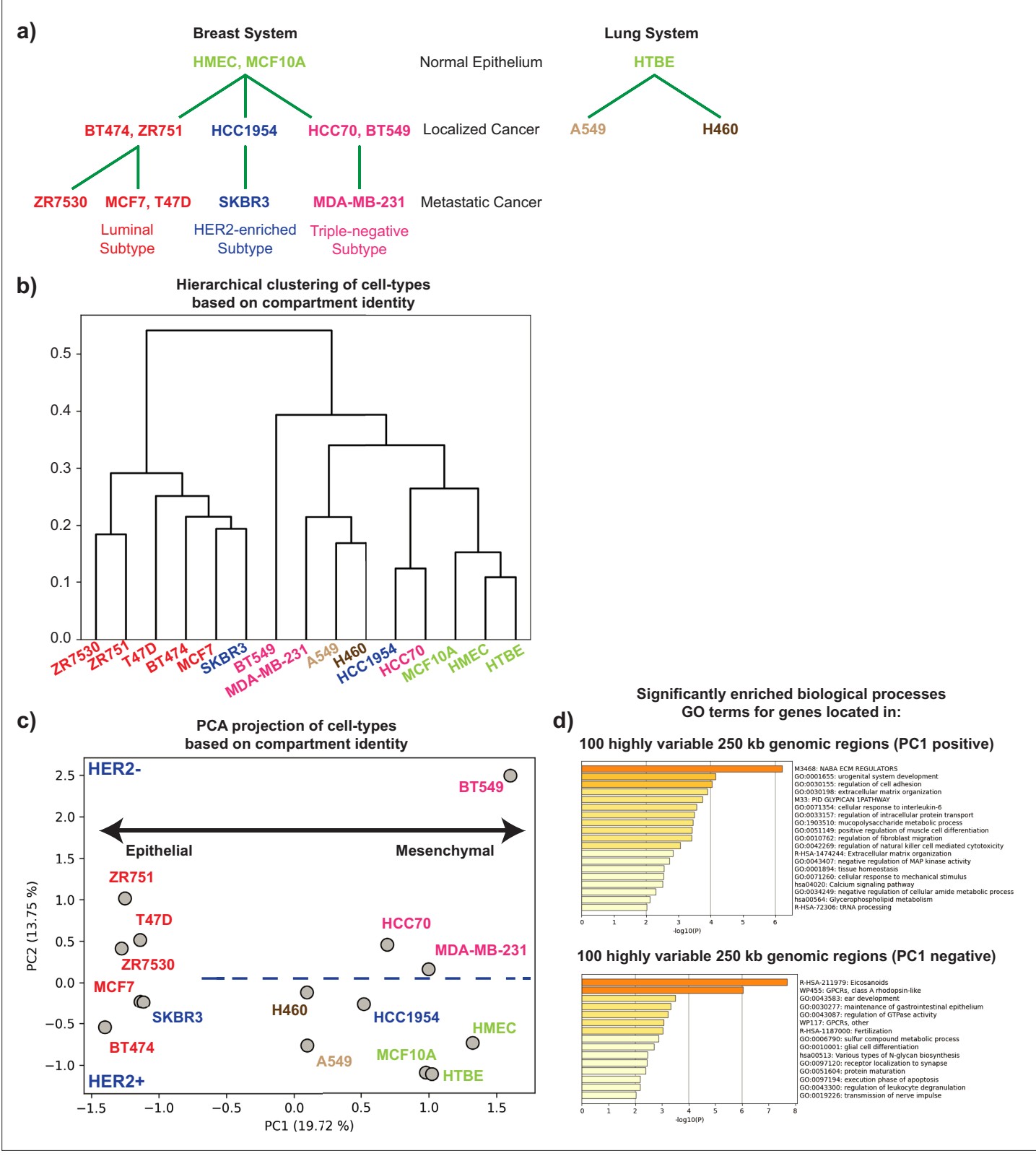

**Figure 1.** Spatial compartment identity segregates cells by subtype and epithelial–mesenchymal status. (**a**) Primary cells and cell lines used in this study are arranged depending on their molecular subtypes and malignancy status based on prior available characteristics. Green: normal breast and lung epithelium, red: luminal subtype, blue: HER2-enriched subtype, pink: triple-negative subtype, brown: localized lung cancer. (**b**) Hierarchical clustering of genome-wide compartment identity data for all breast and lung cell types at 250 kb resolution; colors as in 1 a. (**c**) Principal component analysis (PCA) of

*Figure 1 continued on next page*

*Figure 1 continued*
genome-wide compartment identity data for breast and lung cell types at 250 kb resolution; colors as in (a). The PC2 axis is divided into two subspaces that mostly segregate cell lines based on HER2 status (blue thick dotted line). (**d**) GO term enrichment (biological processes) of the genes in genomic regions corresponding to the top 100 positive and 100 negative elements of the first eigenvector of genome-wide compartment identity PCA.

lines along PC1 from spatial compartmentalization matches quite well with the previously characterized epithelial–mesenchymal classification of these cell lines based on transcriptomic data (*Figure 2— figure supplement 1*; *Le et al., 2018*; *Tan et al., 2014*). We noticed that the PC2 axis tends to segregate cell lines according to the presence or absence of HER2. As expected for an unsupervised analysis like PCA in a complex system, the separation is not perfect, as HER2-negative cell lines like MCF7 are plotted near HER2-positive cells. But the trend of separation suggests that in breast cancer cell lines, the presence or absence of this receptor and its downstream signaling network has important effects on the genome organization at the compartment scale. To further characterize the genomic regions that showed distinctive compartment signatures across cell lines, we extracted lists of all genes that fell into genomic bins with the 200 most extreme eigenvector loadings for PC1: 100 strongest positive and 100 strongest negative bins. Gene ontology enrichment results for these gene lists coincide with the idea that PC1 represents the EMT axis: PC1 positive genes have functions related to cell adhesion (tenascin C, JAG1, ACTA2), extracellular matrix organization (APP, LAMA2, FGF2, BMP4) and positive regulation of cell migration, consistent with a mesenchymal phenotype, while PC1 negative genes show GO terms related to epithelial maturation (*Figure 1d*) Overall, our analyses show that the genome organization is not only able to distinguish different subtypes of cancers but also exhibits changes along the EMT.

## Breast cancer transcriptome profiles also exhibit epithelial–mesenchymal state-based segregation

We next analyzed the gene expression signatures that emerged across different subtypes of breast cancer (data sources described in *Supplementary file 1*) and compared those with the patterns obtained from the compartmental analysis. We first performed PCA on the transcriptome profile of all the breast and lung system cells considered in this study (*Figure 2a*). From the low-dimensional PC projection result, we observed that similar breast cancer subtype cells distribute in the PC space based on their transcriptome profile. As in the compartment signature, MDA-MB-231 cells localize close to lung cancer cells A549 and H460, suggesting that this lung metastatic cell line has transcriptional similarities to lung cancer. We also noticed that along the largest PC axis (PC1), cells are ordered based on their EMT state. Performing GO term enrichment analysis using the genes from the regions with the top 100 most positive and 100 most negative eigenvector loadings for this PC1, we find that genes associated with PC1 positive values relate to epithelial organization, differentiation, and negative regulation of cell migration, denoting a more epithelial phenotype (*Figure 2b*). In contrast, PC1 negative genes relate to the mesenchymal functions with enriched terms for wound healing, hemostasis, cell movement, and ECM remodeling (*Figure 2b*). Along with the unsupervised PCA, we also performed gene expression analysis in a supervised fashion using two different methods – gene set variation analysis (GSVA) and non-negative principal component analysis (nnPCA) (*Figure 2c, d*, see Methods).

Briefly, for each cell type, an E (epithelial) score and an M (mesenchymal) score were calculated based on the expression levels of a curated set of 232 epithelial and 193 mesenchymal genes (*Tan et al., 2014*; *Panchy et al., 2022*). A higher E score represents more epithelial characteristics, whereas a higher M score corresponds to a mesenchymal phenotype. The GSVA and nnPCA transcriptome analyses recapitulate a similar ordering of cell lines according to epithelial–mesenchymal characteristics.

## Distinct EMT gene sets are affected by spatial compartment changes vs. transcription changes, revealing multiple layers of EMT signature

Numerous studies have already demonstrated that cancer cells segregate along an EMT axis by transcriptomics (*Le et al., 2018*; *Tan et al., 2014*; *Panchy et al., 2022*; *Jung et al., 2020*; *Cook and Vanderhyden, 2022*; *Nguyen et al., 2018*). Given that spatial compartmentalization is often correlated with gene expression (*Li et al., 2024*), a simple possible explanation for the EMT segregation of cells by

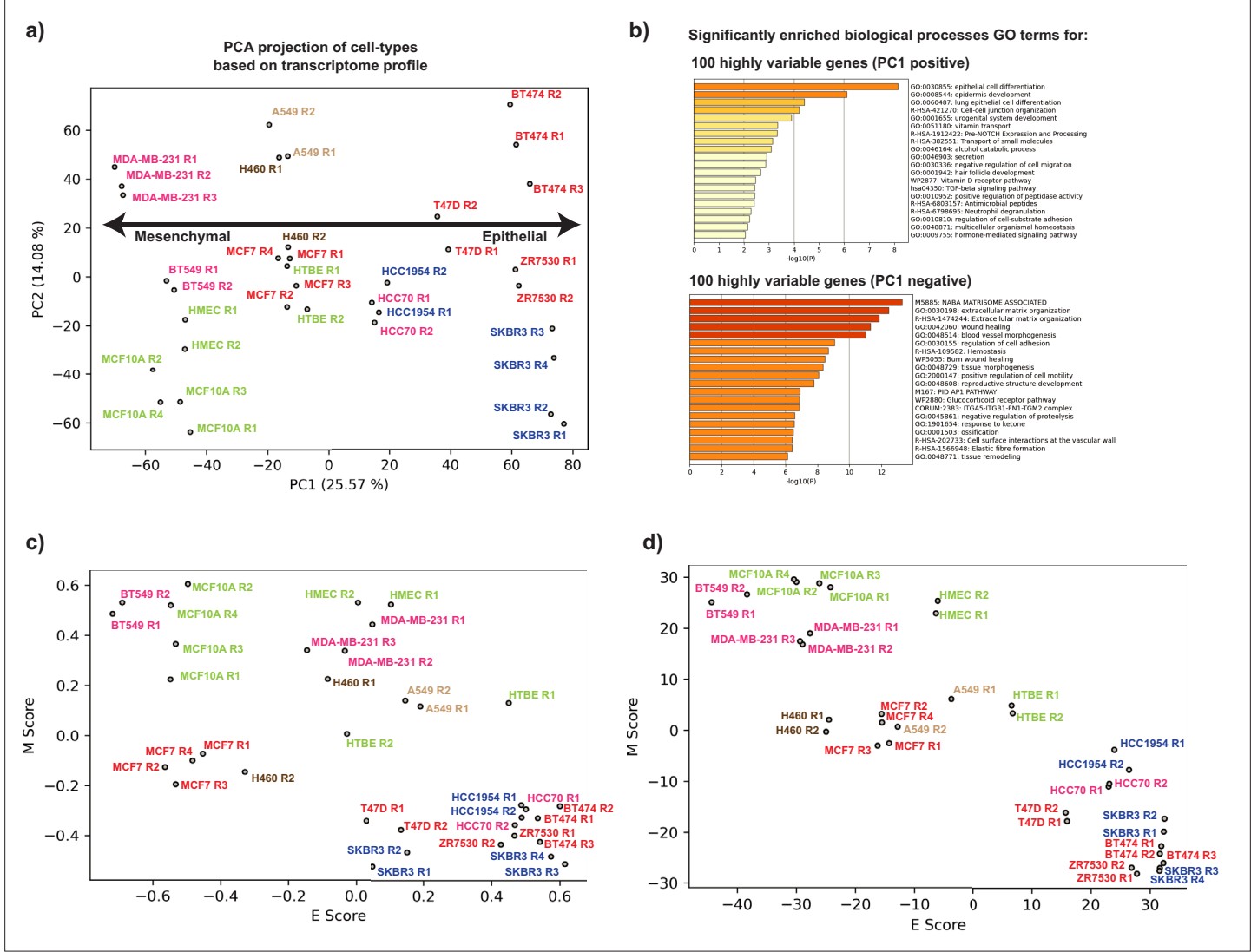

**Figure 2.** Gene expression differences also segregate breast cancer cell lines along an epithelial–mesenchymal transition (EMT) axis. (**a**) Principal component analysis (PCA) of transcriptome data for breast and lung cell types in this study. Cell types are colored based on their molecular subtypes as represented in *Figure 1a*. (**b**) GO term enrichment (biological processes) of the genes corresponding to top 100 positive and 100 negative elements of the first eigenvector of the transcriptome profile PCA. Projection of breast and lung cell types on a curated epithelial and mesenchymal axis based on their gene expression using gene set variation analysis (GSVA) (**c**) and non-negative principal component analysis (nnPCA) (**d**) methods.

The online version of this article includes the following figure supplement(s) for figure 2:

**Figure supplement 1.** Concordance between epithelial–mesenchymal transition (EMT) classification of cell types by gene expression and Hi-C compartments.

compartment signatures could be that genes which change expression also change compartment. An alternative hypothesis would be that transcription and 3D genome organization change represent two independent aspects of the EMT transition. To evaluate these possibilities, we compared the genes obtained from the compartmental PCA analysis, the transcriptome PCA analysis, and the curated epithelial–mesenchymal gene set (*Figure 3a*, gene lists found in *Supplementary file 2*). We found that the genes in regions that change spatial compartmentalization have very little overlap with the sets of genes that most varied in transcription across these cell lines or were curated as differently expressed EMT genes (*Figure 3a*). To further investigate this result, we examined the transcription status of genes that changed compartment across the EMT spectrum and, conversely, the compartment status of genes that changed transcription (*Figure 3b–d*). As expected, hierarchical clustering of the detected highly varying compartments (*Figure 3b*, top) and highly varying transcripts

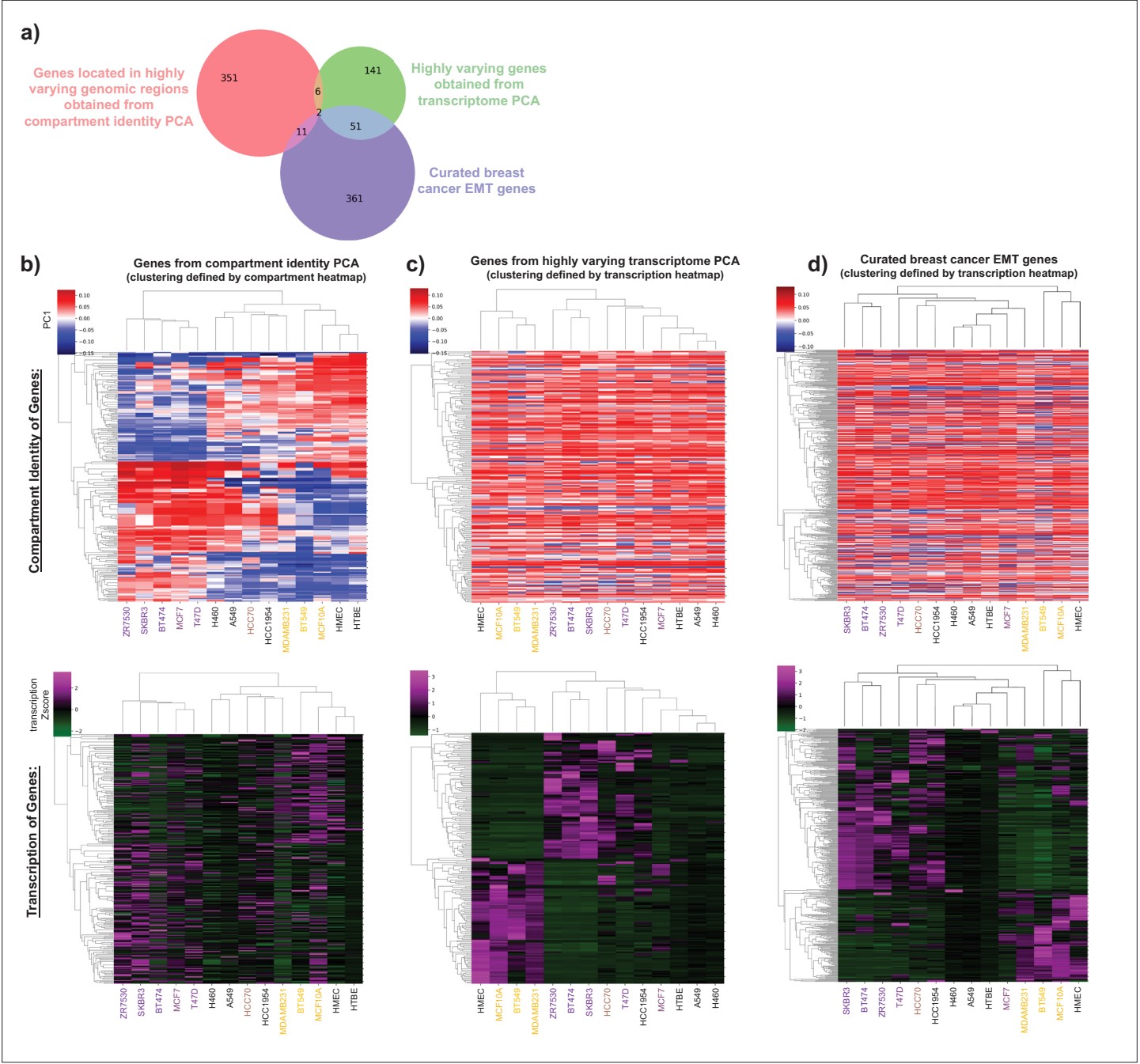

**Figure 3.** Transcription and compartment changes capture distinct sets of epithelial–mesenchymal transition (EMT)-related genomic regions. (**a**) Overlap analysis of the genes obtained from the compartmental analysis PC1 (red, same genes analyzed in *Figure 1d*), the transcriptome analysis PC1 (green, same genes analyzed in *Figure 2b*), and a curated breast cancer epithelial–mesenchymal gene set (blue). See *Supplementary file 2* for gene lists. Hierarchical clustering of compartment profiles (top row) and gene expression (bottom row) of the genes obtained from either (**b**) the compartmental analysis PC1, (**c**) transcriptome analysis PC1, or (**d**) the curated set of breast cancer epithelial and mesenchymal genes. For (**b**), the clustering order is determined by the compartment profile and then the same order of genes and cell lines is shown for the expression data. For (**c**) and (**d**), the clustering order is based on gene expression profiles and then the same order is shown for compartment profiles. Cell line names are colored according to position along the EMT axis (purple = more epithelial, yellow = more mesenchymal) as presented in *Le et al., 2018* and shown in *Figure 2—figure supplement 1a*.

The online version of this article includes the following figure supplement(s) for figure 3:

**Figure supplement 1.** Conservation vs. change in breast cancer cell compartment identity for regions that differ between breast and lung epithelial cells.

(*Figure 3c, d*, bottom) resulted in clear segregation of the cell types into similar groupings (i.e., HMEC and MCF10A together, BT474 and SKBR3 together with an opposing pattern). But the genes which strongly changed compartments between cell lines showed no consistent patterns of correlated expression change (*Figure 3b*, bottom). Likewise, very few compartment differences existed across cell lines at genes whose transcription segregated cells by EMT status (*Figure 3c, d*, top). Notably, for both the transcriptome PCA analysis and the curated EMT gene set, the majority of the differentially expressed genes are found in the active A compartment. Overall, our results favor the hypothesis that compartmentalization and gene expression signatures are capturing two independent factors in EMT transition. The genes that differ most in expression across these cell lines are largely already accessible (A compartment) and therefore their expression is likely not controlled by spatial compartmentalization. Meanwhile, the additional EMT relevant genes that change compartment, but not transcription, may be now spatially poised for future gene regulation given additional environmental cues, as has been shown in other contexts (*Das et al., 2023*; *Stik et al., 2020*; *Leemans et al., 2019*; *Honda et al., 2022*). Overall, the set of genes that changes compartments does not have as clear epithelial and mesenchymal functional enrichment as the transcription change set of genes (as shown by the analysis of these gene sets presented in *Figures 1d and 2b*). This could indicate that some of the compartment changes that occur with EMT are not directly gene regulatory but rather enable an overall conformational change of the chromatin that is needed for the alterations in physical cell state or to accomplish long-distance gene regulation changes. Indeed, we observe specific cases where a compartment change region is located near but not at a gene expression change region (*Figure 2—figure supplement 1c*). For example, TFF3 is in the set of genes that decreases in expression in M vs. E type breast cancer cell lines. While the gene itself remains in the A compartment in most all cell types, it lies near several regions that switch from A to B in M vs. E type cancer cell lines. Therefore, some of the discordant gene expression and compartment change regions may reflect the impact of compartment shifts on gene expression in neighboring regions. Notably, this same genomic region on chr21 showed striking compartment switches with cancer progression in prostate cancer as well (*San Martin et al., 2022*).

## Lung metastatic breast cancer cell lines acquire lung-like genome architecture

To survive a metastatic environment, a cancer cell must express a repertoire of binding factors, extracellular matrix proteins, and ligands that positively interact with the secondary tissue microenvironment and foster colonization and growth. We hypothesized that for that to happen, the genome organization of the metastatic cell should change to resemble that of the cells at the secondary site, as a prelude to transcriptional activation. To evaluate this hypothesis, we compared the 3D genome organization of the metastatic and localized breast cancer cells with lung epithelial vs. lung cancer cells (*Figure 4a, b*). As a baseline, we first compared the breast cancer compartment identity data with that of the normal breast epithelial cells. For each comparison, this resulted in four possible compartment identity combinations: AA (A compartment in both normal breast and breast cancer), AB (A compartment in normal breast and B compartment in breast cancer), BA (B compartment in normal breast and A compartment in breast cancer), and BB (B compartment in both normal breast and breast cancer). Similarly, the compartment identity of lung cancer cell lines was also compared with that of the normal lung cell. We then performed a cross-comparison between the breast and lung cancer comparisons, leading to a total of 16 compartment identity combinations (*Figure 4b*). Of these, we were particularly interested in cases where the compartment state of healthy breast epithelial cells consistently differed from lung. It is notable that this set is relatively small to begin with (on the order of 200 250-kb bins), which emphasizes the overall similarities between epithelial cell types, whether they come from breast and lung. But for these regions that differ between healthy breast and lung, we next asked whether the breast cancer compartment would retain its breast identity or switch to match lung. In all the breast cancer cell lines derived from lung metastases, we found that more than half of these genomic regions switched compartments to match the lung compartment state (*Figure 3—figure supplement 1*). We term these switches that make the breast cancer cell compartment identity more like the lung as 'lung-permissive genome organization changes'. They include: AB_BB (genomic regions which are in the B compartment in both normal lung and localized lung cancer, and which change from A to B in the transition from normal breast to breast cancer) and BA_AA (genomic regions which switch from

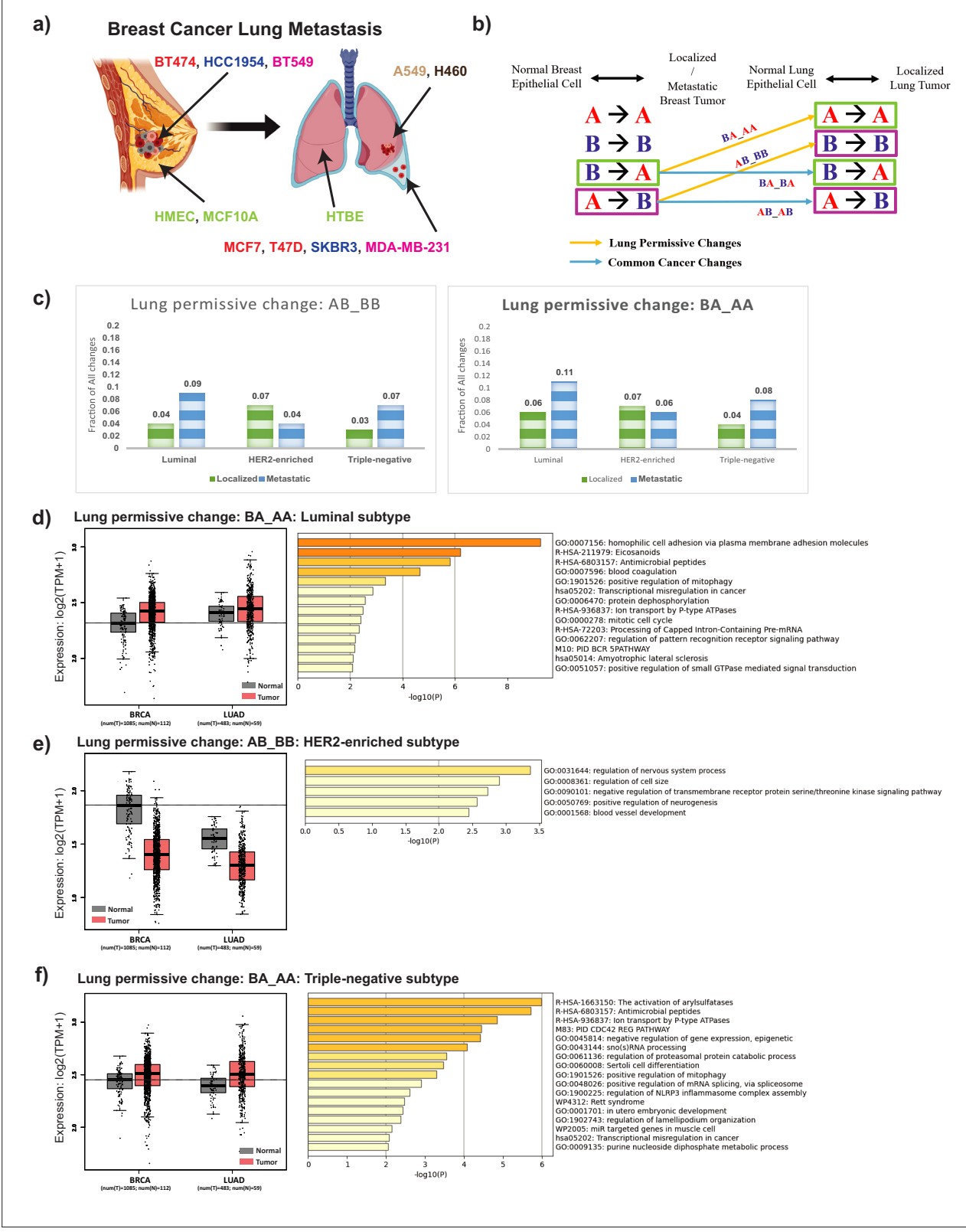

**Figure 4.** Breast cancer cells that metastasize to the lung show increased lung-permissive compartment changes with concordant gene expression alteration in patients. (**a**) Normal breast epithelium (HMEC and MCF10A), localized breast cancer (BT474, HCC1954, and BT549), and lung metastatic breast cancer (MCF7, T47D, SKBR3, and MDA-MB-231) cells are used to model different stages of breast cancer progression. To model diseased lung, normal lung epithelium (HTBE) and localized lung cancer (A549 and H460) cells are used. (**b**) Schematic diagram representing breast cancer lung-

*Figure 4 continued on next page*

*Figure 4 continued*

permissive changes calculation. First, for both breast and lung, cancer cell compartment identity is compared with the normal epithelial cell. This results in four possible compartment identity combinations: AA (A compartment in both normal and cancer), AB (A compartment in normal and B compartment in cancer), etc. Then, a cross-comparison between the breast and lung systems leads to 16 compartment identity combinations. Two specific combinations: AB_BB and BA_AA are defined as lung-permissive changes (yellow arrows). (**c**) Fraction of lung-permissive changes shown by localized and metastatic cancers from different breast cancer subtypes. TCGA patient gene expression (left) and GO term enrichment (biological processes) (right) for genes from regions exhibiting (**d**) BA_AA lung-permissive changes in the luminal metastatic breast cancer subtype, (**e**) AB_BB lung-permissive changes in case of HER2-enriched metastatic breast cancer subtype, or (**f**) BA_AA lung-permissive changes in case of triple-negative metastatic breast cancer subtype. TCGA data from BRCA and LUAD tumor vs. normal sets used and sample size indicated. Points indicate individual samples. Boxes indicate 25th percentile, median, and 75th percentile.

The online version of this article includes the following figure supplement(s) for figure 4:

**Figure supplement 1.** Calculation to adjust organ-permissive compartment switch calculation to account for different background levels of certain compartments.

**Figure supplement 2.** Comparison of brain- and lung-permissive changes in lung metastatic breast cancer.

**Figure supplement 3.** Prostate cancer cells that metastasize to the brain show increased brain-permissive compartment changes at neuronal-related genes.

B to A in breast cancer compared to normal breast epithelium, and which are in the A compartment in both normal lung and lung cancer). These combinations allow us to distinguish changes that could foster lung-specific survival changes from cancer-specific traits such as AB_AB and BA_BA (compartments that change in the same direction from normal to cancer cells in both breast and lung). We then evaluated whether lung-permissive 3D genome changes occurred more frequently in lung metastatic breast cancer than in localized breast cancer.

To quantify the frequency of lung-permissive changes overall, we calculated what fraction of the compartment alterations in breast cancer vs. normal breast tissue were also counted as lung permissive. We divided cells according to breast cancer subtype for this comparison. For the subtypes where several cell lines were available, we considered compartment identity combinations that were consistent between members of the group. We found that for luminal and triple-negative subtypes, metastatic breast cancer exhibits a higher level of lung-permissive changes than localized cancer (*Figure 4c*).

## Genes affected by lung-permissive compartment changes show lung-like expression levels in cancer patients

After selecting the genomic regions that exhibit lung-permissive changes in metastatic breast cancer cell lines, we extracted the underlying genes and used the GEPIA2 tool (*Tang et al., 2017*) to analyze the expression of these gene sets in TCGA breast and lung cancer patient data (*Figure 4d–f*; left panels). We found that in general, the gene expression levels change concordantly with the compartment. The results show that for the regions exhibiting lung-permissive changes, the switch in compartmental identity is associated with the gene expression change from normal to tumor. That is, for BA_AA regions, the breast tumor expression of these genes is higher than normal breast tissue

**Table 1.** Quantifying overlap between epithelial–mesenchymal transition (EMT) and organotropic compartment switched genomic regions.

Each entry represents the fraction of the top 100 positive (left) or negative (right) compartment PC1 regions which are also found in the set of genomic bins that show organotropic compartment switches in the categories shown.

| | PC1 positive 100 significant regions | PC1 negative 100 significant regions |
|---|---|---|
| Luminal primary AB_BB + BA_AA | 0.03 | 0.03 |
| Luminal metastatic AB_BB + BA_AA | 0.03 | 0.03 |
| Her2 primary AB_BB + BA_AA | 0.03 | 0.02 |
| Her2 metastatic AB_BB + BA_AA | 0.03 | 0.02 |
| TNBC primary AB_BB + BA_AA | 0 | 0 |
| TNBC metastatic AB_BB + BA_AA | 0.03 | 0 |

expression, coinciding with higher expression in the lung (*Figure 4d*). Conversely, for AB_BB regions, the breast tumor expression of genes in these regions is lower than for normal tissue, corresponding to lower expression in the lung (*Figure 4e*). Notably, these lung-permissive compartment change genomic regions do not strongly overlap with the key compartment switch regions we identified by principal components analysis (*Table 1*). Correspondingly, the enriched functions of genes in these lung-permissive switch regions (*Figure 4d–f*) are different than the EMT-related functions found among the genes identified by PC1.

## Organotropic 3D genome changes match target organ more than an unrelated organ

To verify that 'lung-permissive changes' truly occurred more than expected in lung metastatic cells, we used a different organ system – brain metastasis and brain cancer – as a negative control. Breast cancer can metastasize to the brain (*Bachmann et al., 2015*; *Yoneda et al., 2001*), but the specific metastatic cell lines for which we have data were not derived from brain metastases. Therefore, we would expect fewer 'brain-like' compartment changes in metastatic breast cancer that metastasizes to lung. Before comparing percent permissive changes between organs, we developed an adjusted version of the calculation to account for differing baseline probabilities of compartment change at the distant organ site (*Figure 4—figure supplement 1* and Methods). The adjusted calculation still results in a higher level of lung-permissive changes in lung metastatic breast cancer cell lines as compared to localized (*Figure 4—figure supplement 2d*). For the brain comparison, we used the NHA cell line (normal human glia) and the SF9427 cell line (glioblastoma) as our reference models (*Figure 4—figure supplement 2a, b*). We then calculated the adjusted level of brain-permissive changes shown by localized and metastatic breast cancers (*Figure 4—figure supplement 2c*). We found that the metastatic breast cancer cells used in this study showed lung-permissive changes more frequently than brain-permissive compartment changes (*Figure 4—figure supplement 2c, d*). This is congruent with the fact that these metastatic breast cancer cell lines were all isolated from lung pleural effusions. We note that Hi-C data for brain metastatic breast cancer cell lines is not currently available for comparison.

## Brain metastatic prostate cancer cell lines exhibit organotropic 3D genome changes toward brain identity

The analysis above suggests that lung metastatic breast cancer cell lines show more lung-permissive than brain-permissive genome organization changes. Next, we asked whether a cancer type that metastasizes to the brain exhibits brain-permissive compartment changes. For this purpose, we chose prostate cancer cell lines derived from localized (lymph node) and brain metastatic sites (LNCaP and DU145) and compared their genome architecture to the non-tumorigenic prostate epithelial cell line RWPE (*Figure 4—figure supplement 3a, b*). We found that the brain metastatic cells showed more brain-permissive changes than the one derived from a lymph node (*Figure 4—figure supplement 3c*). Also, particularly in the B to A direction, there were greater brain-permissive changes in this prostate cancer system (*Figure 4—figure supplement 3c*) than in the breast cancer system (*Figure 4—figure supplement 3c*). This confirms that a cancer type that metastasized to the brain matched the brain genome compartmentalization profile more often than a cancer type that had not metastasized to the brain. We found that the genes in the switched compartment regions have functions that could relate to leaving the prostate epithelium and adapting to the brain environment (*Figure 4—figure supplement 3d*). For example, the DU145 (brain metastatic) brain-permissive B to A switched genes were enriched for functions in regulation of synapse organization (including calcium channel CACNA2D3). Conversely, the regions that switch from A to B to match the brain B compartment and are enriched for keratinization and cell adhesion, suggesting a downregulation of prostatic epithelial properties.

## Discussion

During metastasis, breast cancer cells spread preferentially to certain organs, a series of events widely known as metastasis organotropism. This phenomenon, also shown by other types of cancer, may be explained, to some extent, by the 'seed and soil' hypothesis which states that cancer cells will be able to colonize a distant site if the cells have intrinsic ability to survive in the microenvironment of the secondary site. Though the process of metastasis is complex and involves other factors, the

primary tumor microenvironment conveys survival capabilities to cancer cells that can potentially favor survival in other organs; for this to occur, the cancer cell must engage specific gene expression signatures. However, there are many open questions related to the rise of organ-specific survival traits. Are these gene signatures a result or consequence of genome organization changes? What is the role of genome rearrangements in metastasis organotropism for which there are no significant gene expression alterations? Does the genome organization of the metastatic cell become similar to that of the cells at the distant site? In this work, we have made a step toward answering these questions by analyzing compartmental organization data of breast cancer cell lines derived from lung metastasis in comparison to lung cell lines.

Unsupervised clustering of compartment organization data showed that compartment organization can distinguish cancer cell lines by subtype, providing the first indication that lung metastatic cell lines may show similarities to lung cell genome organization. For example, the triple-negative invasive metastatic breast cancer cell MDA-MB-231 clusters close to the lung carcinoma cells (A549 and H460). The clustering result supports the conclusion that the breast cancer subtypes are not only quite different at the gene expression level, but also at the level of genome organization. Furthermore, it also shows that the genome organization is highly consistent among luminal subtype cancers, irrespective of the tissue from which the cells were collected, while the other subtypes, such as triple-negative, were more heterogeneous in their structure. This is congruent with studies involving gene expression which have shown that the triple-negative breast cancer category is highly heterogeneous, encompassing six subtypes (*Wang et al., 2019a*).

When we compared the compartmental organization changes shown by the localized and lung metastatic breast cancers to the compartment state of the lung, we found that, for luminal and triple-negative subtypes, metastatic cells exhibit higher percentages of lung-permissive changes compared to localized cancers. Given that these cancer cell lines were derived from lung pleural effusions, we cannot tell whether these metastatic cells acquired a lung-like compartmental organization at the primary tumor site or later while adapting to the metastatic site. To answer this, we would need temporal genome organization data of metastatic progression, which is not currently available. Our current study is also limited by the use of cell lines derived from multiple different patient sources and grown in culture. To validate and clarify our observations, future research would need to analyze the 3D genome structure of matched primary and metastatic tumors from the same human patient.

For the lung-permissive chromatin conformation changes, we find that shifts toward the A compartment are favored compared to the B compartment. It may be possible that shifts toward the A compartment are predominant because both activation and gene repression are possible once a region is in the A compartment, while a shift to the B compartment is not necessary to accomplish gene repression (*Das et al., 2023*; *Hakim et al., 2011*; *Vieux-Rochas et al., 2015*). Alternatively, this preference for shifts toward an A compartment could also be related to the general loss of heterochromatin previously observed in cancer (*Carone and Lawrence, 2013*). Indeed, we found that genes in these spatially reorganized regions, in patient samples, tend to change expression concordantly with their compartment changes. Overall, our results are consistent with the idea that genome compartmentalization can play a role in facilitating organ-specific metastasis.

While our gene expression and compartment analyses echo the current dogma that 3D genome compartments correlate with gene expression changes, further analysis revealed that gene expression and compartment signatures across breast cancer types surprisingly capture two orthogonal sets of genomic regions. We found that compartment profiles group breast cancer cell lines according to their epithelial vs. mesenchymal states (*D'Amato et al., 2012*). This observation becomes much more visible in the PCA transformed data as the cells are arranged along an epithelial–mesenchymal spectrum. Pathways of EMT such as the loss of epithelial identity and the gain of migratory ability can play important roles in facilitating metastasis, though it is important to note that EMT is only one of many complex processes involved in metastasis (*Celià-Terrassa and Kang, 2024*). While epithelial vs. mesenchymal states have been clearly associated with gene expression, it is also becoming clear that these states exhibit characteristic genome architectures as well. Indeed, a recent study shows that genome organization changes during EMT in ovarian cancer systems, by considering some representative epithelial and mesenchymal cell lines (*Pang et al., 2022*). Here, we extend this conclusion for another major cancer, the breast cancer system, by including a spectrum of breast and lung cancer and normal cell lines. Across this cohort, we find not only that both 3D genome architecture and

gene expression changes occur during EMT, but also that these changes happen in different regions. The 3D genome changes that most strongly capture the differences across the epithelial and mesenchymal spectrum occur at regions where gene expression is not notably distinct between epithelial and mesenchymal cells. Likewise, the signature gene expression changes occur in regions that tend to be in the A compartment across all cell types. This suggests that many genes key to EMT are regulated by local factors in the A compartment, while major spatial shifts in chromosome regions may alter long-distance regulation of genes in other regions, change the likelihood of future inducibility of genes, or represent alterations in the physical/mechanical properties of the cell during EMT, which can affect chromatin state (*Wang et al., 2019b*; *Scott et al., 2023*). Overall, our results suggest that genome spatial compartment changes can help encode cancer cell states that may play a role in metastasis as well as enabling survival in a new organ context.

## Methods

### Hi-C data processing

To process Hi-C sequencing reads, we used the HiC-Pro (https://github.com/nservant/HiC-Pro; *Servant, 2022*) pipeline with default parameters (*Servant et al., 2015*). The reads were aligned to the hg19 reference genome. Also, for each cell type, all the available reads from all replicates were merged together to improve the quality of contact data for further downstream processing. After obtaining the valid pair interactions, we binned the interactions at 250 kb resolution for each individual chromosome and normalized the matrices using the ICE technique (*Imakaev et al., 2012*). Following that, the number of normalized contacts of each chromosome was scaled to 1M using 'scaleMatrix.pl' script from the cworld-dekker (https://github.com/dekkerlab/cworld-dekker; *Lajoie, 2019*) tool suite. And finally, in order to get the compartmental organization for each individual chromosome, principal components analysis was performed on the normalized and scaled matrices using the 'matrix2compartment.pl' script from the cworld-dekker tool suite. The compartment analysis assigns A/B compartment identity to genomic regions based on the sign of the elements of the eigenvector (A-positive and B-negative).

### RNA-seq data processing

The RNA-seq files were processed as per the steps described in the previous publication (*Das et al., 2023*). Briefly, the BBDuk (https://github.com/kbaseapps/BBTools; *Riehl et al., 2025*) tool was used first for adapter and quality trimming (*Bushnell et al., 2017*). After that, the reads were mapped to hg19 human reference genome using STAR aligner (RRID:SCR_004463; *Dobin et al., 2013*). Once aligned reads were generated, HTSeq-Counts (https://github.com/simon-anders/htseq; *Anders and Zanini, 2020*) tool was used to perform gene level feature count (*Anders et al., 2015*). As the RNA-seq files were obtained from various sources and generated using different experimental techniques, they suffer from the batch effect which masks the biological signal. To remove the batch effect, ComBat-seq (https://github.com/zhangyuqing/ComBat-seq; *Zhang, 2020*) tool was applied on the raw gene counts data (*Zhang et al., 2020*).

### Hierarchical clustering of genome-wide compartmental organization

To cluster different cell types based on their compartmental organization, hierarchical clustering was performed on the genome-wide compartment identity data. First, the pairwise similarity score between all the cells was calculated using Pearson's correlation. The similarity data was then further converted to distance data by subtracting from 1. And finally, hierarchical clustering was performed on the distance data using 'ward' linkage.

### Gene set variation analysis

GSVA was performed to compute the enrichment scores for individual cell lines using RNA-seq data (*Deshmukh et al., 2021*; *Hänzelmann et al., 2013*). A list of 232 epithelial signature genes and a list of 193 mesenchymal signature genes were used to compute the E enrichment score (E score) and M enrichment score (M score), respectively (*Tan et al., 2014*; *Panchy et al., 2022*; *Cursons et al., 2018*).

### Non-negative principal component analysis

We performed nnPCA using the gene sets mentioned above. nnPCA provided higher resolutions for large number of samples in the EMT spectrum compared to GSVA according to previous studies

(*Panchy et al., 2022*; *Panchy et al., 2021*). nnPCA determines the approximately orthogonal axes with non-negative coefficients (loadings) for features (genes). Variances of projections of data points (cell lines) onto these axes are maximized via an optimization method (*Sigg and Buhmann, 2008*).

## Correction for comparing organ-permissive changes across different organs

It is possible that while comparing the host organ-permissive changes across different secondary organs, the compartment switch ratio of different secondary localized cancers might include inherent bias to the comparison. To eliminate the bias, for the secondary organ systems, we first calculate the probability of a genomic region to be in a specific compartment in localized cancer given that the region belongs to the same compartment in normal cell (*Figure 4—figure supplement 1*). For the A compartment, p(TA|NA) represents the probability of a region in the A compartment in localized cancer cells given that the region also belongs to the A compartment in normal cells. Similarly, p(TB|NB) for the B compartment. If we have more than one cell line representing the secondary site localized cancer, the minimum probability across all the cells is considered. Once we have those, the probabilities are subtracted from one and the resultant values are multiplied with corresponding secondary organ-specific percentages. For example, (1 − p(TA|NA)) is multiplied to the percentage of BA_AA and (1 − p(TB|NB)) to the AB_BB.

## Acknowledgements

The authors would like to thank all the members of the McCord lab for their fruitful suggestions in developing the research project. This work was supported by the National Institutes of Health NIGMS grant R35GM133557 to RPM and American Cancer Society postdoctoral fellowship 134060-PF-19-183-01-CSM to RSM. TH is supported by the National Institutes of Health NIGMS grant R35GM149531. This work used Bridges2 at Pittsburgh Supercomputing Center and Expanse at San Diego Supercomputer Center through allocation BIO230105 (awarded to PD) from the Advanced Cyberinfrastructure Coordination Ecosystem: Services & Support (ACCESS) program, which is supported by National Science Foundation grants #2138259, #2138286, #2138307, #2137603, and #2138296.

## Additional information

### Funding

| Funder | Grant reference number | Author |
|---|---|---|
| National Institute of General Medical Sciences | R35GM133557 | Rachel Patton McCord |
| American Cancer Society | 134060-PF-19-183-01-CSM | Rebeca San Martin |
| National Institute of General Medical Sciences | R35GM149531 | Tian Hong |

The funders had no role in study design, data collection, and interpretation, or the decision to submit the work for publication.

### Author contributions

Priyojit Das, Conceptualization, Data curation, Formal analysis, Investigation, Visualization, Methodology, Writing – original draft, Writing – review and editing; Rebeca San Martin, Conceptualization, Data curation, Formal analysis, Writing – original draft, Writing – review and editing; Tian Hong, Data curation, Formal analysis, Writing – review and editing; Rachel Patton McCord, Conceptualization, Supervision, Funding acquisition, Visualization

### Author ORCIDs

Priyojit Das ⓘ https://orcid.org/0000-0002-6774-6718
Rebeca San Martin ⓘ https://orcid.org/0000-0001-7249-3922
Tian Hong ⓘ https://orcid.org/0000-0002-8212-7050

Rachel Patton McCord [iD] https://orcid.org/0000-0003-0010-5323

Reviewer #1 (Public review): https://doi.org/10.7554/eLife.103697.3.sa1
Reviewer #2 (Public review): https://doi.org/10.7554/eLife.103697.3.sa2
Author response https://doi.org/10.7554/eLife.103697.3.sa3

## Additional files

### Supplementary files

Supplementary file 1. Classifications of cell line models used and sources of all publicly available Hi-C and RNA-seq data used in the analyses. Information about cell lines derived from the following sources: ATCC, *Engel and Young, 1978*, *Gazdar et al., 1998*, Cellosaurus.org, *Wang et al., 2021*, and *Stone et al., 1978*.

Supplementary file 2. Lists of genes obtained from compartmental PCA, transcriptomics PCA, and curated epithelial–mesenchymal gene set. Compartmental_PC1_Positive: genes found within the top 100 highly varying genomic regions in compartmental PCA analysis (positive PC1); Compartmental_PC1_Negative: genes found within the top 100 highly varying genomic regions in compartmental PCA analysis (negative PC1); RNAseq_PC1_Positive: top 100 highly varying genes in transcriptomics PCA analysis (positive PC1); RNAseq_PC1_Negative: top 100 highly varying genes in transcriptomics PCA analysis (negative PC1); Curated_Epithelial: curated epithelial genes; Curated_Mesenchymal: curated mesenchymal genes.

MDAR checklist

### Data availability

Datasets used for this project were obtained from NCBI GEO, EMBL ENA, 4DN Data Portal, and ENCODE repositories and their accession IDs are given in *Supplementary file 1*. Published software used to process the RNA-seq and Hi-C datasets are cited in the Methods section along with URLs where these software are available.

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
