## [Editor Report · eLife Assessment]

This **valuable** study explores the role of spatial genome organization in oncogenic transformation, addressing an ambitious and significant topic. The authors have assembled comprehensive datasets from various subtypes of localized and lung-metastatic breast cancer cells, as well as from healthy and cancerous lung cells. They identified switching patterns in the 3D genome organization of lung-metastatic breast cancer cells, revealing a reconfiguration of genome architecture that resembles that of lung cells. This provides **solid** evidence with significant biomedical implications for epigenetic regulation in both normal physiology and disease.

---

## [Referee Report · Reviewer #1 (Public review)]

Summary:

This study utilized publicly available Hi-C data to ensemble a comprehensive set of breast cancer cell lines (luminal, Her2+, TNBC) with varying metastatic features to answer whether breast cancer cells would acquire organ-specific feature at the 3D genome level to metastasize to that specific organ. The authors focused on lung metastasis and included several controls as the comparison including normal mammary lines, normal lung epithelial lines, and lung cancer cell lines. Due to the lower resolution at 250KB binning size, the authors only addressed the compartments (A for active compartment and B for inactive compartment) not the other 3D organization of the genome. They started by performing clustering and PCA analysis for the compartment identity and discovered that this panel of cell lines could be well separated based on Her2 and epithelial-mesenchymal features according to the compartment identity. While correlating with the transcriptomic changes, the authors noticed the existence of concordance and divergence between the compartment changes and transcriptomic changes. The authors then switched gear to tackle the core question in metastatic organotropism to the lung. They discovered a set of "lung permissive compartment changes" and concluded that "lung metastatic breast cancer cell lines acquire lung-like genome architecture" and "organotropic 3D genome changes match target organ more than an unrelated organ". To prove the latter point, the authors enlisted additional non-breast cancer cell line (prostate cancer) in the setting of brain metastasis. This is a piece of pure dry computational work without wet bench experiments.

Strengths:

The authors embarked on an ambitious journey to seek for the answer regarding 3D genome changes predisposing metastatic organotropism. The authors succeeded in the assembly of a comprehensive panel of breast cancer cell lines and the aggregation of the 3D genome structure data to conduct a hypothesis driven computation analysis. The authors also achieved in including proper controls representing normal non-cancerous epithelium and the end organ of interest. The authors did well in the citation of relevant references in 3D genome organization and EMT.

---

## [Referee Report · Reviewer #2 (Public review)]

Summary:

This work addresses an important question of chromosome architecture changes associated with organotopic metastatic traits, showing important trends in genome reorganization. The most important observation is that 3D genome changes consistent with adaptations for new microenvironment, including lung metastatic breast cells exhibiting signatures of the genome architecture typical to a lung cell-like conformation and brain metastatic prostate cancer cells showing compartment shifts toward a brain-like state.

Strengths:

This work presents interesting original results, which will be important for future studies and biomedical implications of epigenetic regulation in norm and pathology.

---

## [Author Response]

The following is the authors’ response to the original reviews

**Public Reviews:**

**Reviewer #1 (Public review):**
Strengths:The authors embarked on an ambitious journey to seek the answer regarding 3D genome changes predisposing to metastatic organotropism. The authors succeeded in the assembly of a comprehensive panel of breast cancer cell lines and the aggregation of the 3D genome structure data to conduct a hypothesis-driven computation analysis. The authors also achieved in including proper controls representing normal non-cancerous epithelium and the end organ of interest. The authors did well in the citation of relevant references in 3D genome organization and EMT.Weaknesses:(1) The authors should clearly indicate how they determine the patterns of spread of the breast cancer cell lines being utilized in this manuscript. How did the authors arrive at the conclusion that certain cell lines would be determined as "localized spread" and "metastatic tropism to the lung"? This definition is crucial, and I will explain why.

It is indeed a critical point to clearly define and explain what qualifies as metastatic potential to particular organs in our system. Here, we intentionally limited our scope to metastasis that had occurred within the human system. Our cell lines are chosen based on their sites of origin and etiological history in the patients from which they were derived. For example, the cancer cell line BT474 was classified as “localized” because these cells were derived from a solid tumor in the breast itself. Meanwhile, MCF7 and T47D cell lines are considered lung metastatic because these cells were collected from the pleural effusion from the lung. We therefore model human organotropism from the breast to the lung by using cells that originated from infiltrative ductal carcinoma (human breast) but were collected from pleural effusions (human lung). We then use as a comparison a human lung cancer-derived cell line that was itself purified from a pleural effusion. In this way, we can compare the genome structure of a lung cancer cell in the lung environment to a breast cancer cell that has metastasized to the lung environment.

In our revised version, we further clarify this definition in the text as well as in additional annotations in our supplemental table of all cell line information.

Todd Golub's team from the Broad Institute of MIT and Harvard published "A metastasis map of human cancer cell lines" to exhaustively create a first-generation metastasis map (MetMap) that reveals organspecific patterns of metastasis. (By the way, this work was not cited in the reference in this manuscript.) The MetMap Explorer (https://depmap.org/metmap/vis-app/index.html) is a public resource that could be openly accessed to visualize the metastatic potential of each cell line as determined by the in vivo barcoding approach as described in the MetMap paper in the format of petal plots. 5 organs were tested in the MetMap paper, including brain, lung, liver, kidney, and bone. The authors would discover that some of the organ-specific metastasis patterns defined in the MetMap Explorer would be different from the authors' classification. For example, the authors defined MCF7 as a line as lung metastatic, and rightly so the MetMap charted a signal towards lung with low penetrance and low metastatic potential. The authors defined ZR751 as a line with localized spread, however, the MetMap charted a signal towards the kidney with low penetrance and low metastatic potential, the signal strength similar to the lung metastasis in MCF7. A similar argument could be made for T47D. The TNBC line MDA-MB-231 is indeed highly metastatic, however, in MetMap data, its metastasis is not only specific to the lung but towards all 5 organs with high penetrance and metastatic potential. The 2 lung cancer cell lines mentioned in this study, A549 and H460, the authors defined them as localized spread to the lung. However, the MetMap data clearly indicated that A549 and H460 are highly metastatic to all 5 organs with high penetrance and high metastatic potential.

We acknowledge the valuable contributions of animal models in metastatic cancer studies, but we also want to avoid the potentially confounding variable of the animal microenvironment. The MetMap Explorer contains valuable information (and as part of our clarification on this point, we now cite the MetMap in the text), but the “metastatic potential of each cell line” for this tool is measured in a mouse environment. Knowing that a particular cell line, which originated from a human lung metastasis, can further metastasize to other organs in a mouse does not necessarily mean that those cells could do so in humans. The microenvironment responses to metastatic colonization recapitulate the events in wound repair, and these can differ among species (https://pubmed.ncbi.nlm.nih.gov/28916657/
https://pubmed.ncbi.nlm.nih.gov/39729995/). Further, the changes a cell needs to make to adapt to a new organ system in a mouse could be confounded by the changes needed to adapt to mouse conditions in general. Finally, migration from a site of ectopic injection may not mimic migration from an initial tumor site. These factors lead to well known cases where MetMap does not reflect the metastatic potential of cancers in humans. As a classic example, prostate cancer frequently metastasizes to bone in humans, and the PC3 cell line was derived from a bone metastatic prostate cancer. However, MetMap shows no evidence of PC3 being able to metastasize to bone in a mouse.

We agree that the very best data would come from matched primary and metastatic tumors in the same human patient, but those data do not currently exist and generating them would require future work beyond the scope of this study.

Since results will vary among different experimental models testing metastatic organotropism, (intracardiac injection was the metastasis model being adopted in the MetMap), the authors should state more clearly which experimental model system served as the basis for their definition of organ-specific metastasis. In my opinion, this is the most crucial first step for this entire study to be sound and solid.

Taking all the above into account, in our revision, we have now included further clarification in the main text to more clearly explain how and why we chose the cell lines we did and what the advantages and limitations of this choice are.

(2) Figure 1b: The authors found that "MDA-MB-231 cells were grouped with the lung carcinoma cells. This implies that the genome organization of this cell line is closer to that of lung cells than to other breast epithelial cell lines.". In fact, another TNBC line BT549 was also clustered under the same clade. So this clade consisted of normal-like and highly metastatic lines. Therefore, the authors should be mindful of the fact that the compartment features might not directly link to metastasis (or even metastatic organotropism).

In figure 1b, the grouping that includes MDA-MB-231 (lung metastatic breast cancer) connected to A549, and H460 (lung cancer) occurs at a distance of about 0.2. If the clustering tree were cut at a distance of 0.26, 6 separate clusters would result: two clusters of Luminal subtypes (all labeled red), one that includes all healthy epithelial cells (both lung and breast, all labeled green), one that links two localized breast cancers, one that links MDA-MB-231 to lung carcinoma cell lines, and then BT549 by itself. So, while BT549 appears next to MDA-MB-231 along the horizontal axis, this is just coincidence of the representation: the dendrogram shows it is quite distant from all the other cell lines in this cluster according to compartment profile.

So, it is only MDA-MB-231 that is very closely linked with the lung cancer cell types.

It is true that the healthy lung cells (HTBE) are clustered separately and are more similar to normal/non tumorigenic breast epithelial cells (HMEC and MCF10A) than to any cancer cell type. This could suggest that there are aspects of the compartment pattern that represent any healthy epithelium as compared to cancer. What we find in the compartment profile, in both the clustering and the PCA analysis, is that compartment signatures contain information about cell properties on several overlapping levels: there is an aspect of the compartment profile that distinguishes healthy from cancerous cells, an aspect that distinguishes luminal cancers from other subtypes, a part that associates with organotropism, and an aspect that captures EMT status. The final compartment status is a composite of these numerous factors.

We have clarified the text to indicate that we mean MDA-MB-231 clusters near lung cancer, not necessarily healthy lung cell models.

(3) Figure 3: In the text, the authors stated, "To further investigate this result, we examined the transcription status of genes that changed compartment across the EMT spectrum and, conversely, the compartment status of genes that changed transcription (Fig. 3b, c, and d)". However, it was not apparent in the figure that the cell lines were arranged according to an EMT spectrum.

To display these comparisons more clearly, we have now revised figure 3b, c, and d in two ways: First, we have defined the gene and cell line clustering by one set of data (for example, compartment identity in 3b) and then displayed the other data (gene expression) with all genes and cell lines in the same order. Therefore, for each column, genes and cell lines can be compared visually between top and bottom rows. Second, we have colored cell line names from purple to yellow according to their EMT scores as shown in Supplementary Figure 1a. This allows a visual indication of how the clustering separates cell lines by EMT status.

Also, the clustering heatmaps did not provide sufficient information regarding the genes with concordant/divergent compartments vs transcription changes. It would be more informative if the authors could spend more effort in annotating these genes/pathways.

We want to clarify that the genes plotted in the heatmaps in Figure 3 are also the genes whose functional enrichment we present in figures 1 and 2. So, the genes that segregate strongly based on A/B compartment (but not gene expression) in figure 3b are the same genes whose GO terms are annotated in Figure 1d. Likewise, the genes that segregate strongly based on gene expression, but not A/B compartment, in figure 3c and d are the same genes whose GO terms are annotated in Figure 2b. We have now made this connection clearer in the text.

But, we also agree with the reviewer that it is important to explore a bit further the relationship between these divergent sets of genes. Our explorations have led to several observations:

(1) In some cases, the compartment-segregated genes and the transcription-segregated genes are different members of the same pathways. In Author response image 1 below, for example, we show interactions (according to STRING) for genes from figure 3c that are highly expressed in the epithelial-like cell lines and are annotated as involved in epithelial development (green). We then added to the network genes from figure 3b that are specifically in the A compartment in the epithelial-like cell lines but not mesenchymal cell lines that are also annotated as involved in epithelial development (red). Most of these epithelial development genes that change expression are in the A compartment in all cell lines and therefore do not rely on spatial compartment changes for their regulation. But some additional epithelial development genes, which are interconnected in this same network, are changing compartments across the EMT spectrum. One example, FOXA1, is a key hub in the network and is known to be a pioneer transcription factor involved in development and differentiation. Controlling this gene at the level of spatial genome organization rather than local transcriptional control could be important in the stable cell fate changes that can happen with EMT.

(2) Overall, the set of genes that change compartments does not have as strong functional enrichment as the transcription change set of genes. This could indicate that some of the compartment changes that occur with EMT are not directly gene regulatory but rather enable an overall conformational change of the chromatin that is needed for the alterations in physical cell state or to accomplish long distance gene regulation changes.

(3) Related to long distance gene regulation changes, we also see cases in which the gene that changes transcription but not compartment across EMT is adjacent to regions that switch compartments.

A good example is TFF3 (yellow, Supplementary figure 1C). TFF3 is one of the genes that strongly segregates across EMT by transcription, being more highly expressed in epithelial-like (bottom 4 tracks) but not mesenchymal-like (top 4 tracks) cancers. Despite this differential expression, it is almost always in the A compartment across all cell lines. However, it is adjacent to regions that show strong compartment change EMT signatures. So, even though this specific gene region is not changing compartment, its regulation may be influenced by the entire region being Aassociated in epithelial-like but neighboring regions becoming B-associated in mesenchymal like cancers.

TFF3 is expressed in normal breast epithelium and has been implicated as a biomarker for endocrine therapy response in breast cancer.

Meanwhile, many genes that are in these compartment switching regions (BACE2, DSCAM, PDE9A) are not among the strongest expression signature genes.

(4) Interestingly, some of the regions (such as the region shown in Supplementary figure 1C) that change compartment across the breast cancer spectrum overlap with regions that we found change compartment in the progression of prostate cancer, as shown in the string.db enrichment analysis below.

**Author response image 2. sa3fig2:** 

In our revised manuscript, we now include more of these explanations in the text and include the example offset compartment and transcription change region shown about as panel c of Supplementary Figure 1.

(4) Figure 4: The title of the subheading of this section was 'Lung metastatic breast cancer cell lines acquire lung-like genome architecture". Echoing my comments in point 1, I am a bit hesitant to term it as "lung metastatic" but rather "metastatic' in general since cell lines such as MDA-MD-231 do metastasize to other organs as well. However, I do get the point that the definition of "lung metastasis" is derived from the common metastasis features among the cell lines here (MCF7, T47D, SKBR3, MDAMB-231). There might be another argument about whether the "lung" carcinoma cell lines can be considered "localized" since they are also capable of metastasizing to other organs.

Rather than classifying cells on metastatic “potential” (as measured in a mouse), our cell lines are chosen based on their sites of origin and etiological history in the patients from which they were derived. Cancer cell lines called “lung metastasis” were collected from the pleural effusion from the human lung. Likewise, we call a cancer “localized” because it was taken from the tissue where the cancer originated, even if it might, if placed into a different context, be able to metastasize. We would argue that the genome structure features of the “localized” cancers reflect cancers that have not yet metastasized (even if they could in the future) while the “metastatic” cancers have already gone to a certain location (even if they could in theory have gone to a different location).

In a way, what the authors probably were trying to leverage here is the "tissue" identity of that organ.Having said this, in addition to showing the "lung permissive changes", the authors should show the "breast identity conservation" as well. Because this section started to deal with the concept of "tissue/lineage identify", the authors should also clarify whether these breast cancer cell lines capable of making lung metastasis are also preserving their original tissue identity from the compartment features (which would most likely be the case).

This is a great question. We have now more explicitly checked the proportions of genomic regions that change compartments to match lung vs. maintaining breast-specific compartment identity. The graphs in Supplementary Figure 2 begin with all genomic bins that have distinctive compartment identity between non-cancerous breast and lung epithelial cells. Then, the plots show what fraction of these tissue-specific bins change compartment to match lung vs. maintaining breast identity in each breast cancer cell line category. As we have shown in other graphs, particularly for switches to the A compartment, more bins change to match lung in the metastatic vs. primary site cell lines. In most cases, more than 50% of the tissue-specific bins shift to look more like lung.

(5) Rest of the sections: The authors started to claim that the organ-specific metastasis permissive compartmental features mimic the destinated end organ. The authors utilized additional non-breast cancer cell lines (prostate cancer cell lines LNCaP as localized and DU145 as brain metastatic) in brain metastasis to strengthen this claim. (DU145 in MetMap again is highly metastatic to lung, brain, and kidney). However, this makes one wonder that for cell lines that are capable of metastasizing to multiple organ sites (eg. MDA-MB-231, DU145, A459, H460), does it mean that they all acquire the permissive features for all these organs? This scenario is clinically relevant in Stage 4 patients who often present with not only one metastatic lesion in one single organ but multiple metastatic lesions in more than one organ (eg. concomitant liver and lung metastasis). Do the authors think that there might be different clones having different tropism-permissive 3D genome features or there might be evolutionary trajectory in this?

In my opinion, to further prove this point, the authors might need to consider doing in vivo experiments to collect paired primary and organ-specific metastatic samples to look at the 3D genome changes.

We agree that an ideal experimental follow up to this study would be to collect paired metastatic and primary tumors, either in mouse xenograft or, even better, from patients. This is beyond the scope of what we can do for our current paper, but we have added a statement to the discussion of further experiments that would be required to clarify this point.

(6) Technically, the study utilized public Hi-C data without generating new Hi-C data. The resolution of the Hi-C data for compartments was set at 250KB as the binning size indicating that the Hi-C data was at lower resolution so it might not be ideal to address other 3D genome architecture changes such as TADs or long-range loops. It is therefore unknown whether there might be permissive TAD/loop changes associated with organotropism and this is the limitation of this study.

Our decision to focus on A/B compartmentalization rather than TAD or loop structure in this analysis was intentional and biologically motivated, rather than solely being a reflection of data resolution. Both compartments and topologically associated domains (TADs) are key parts of genome organization and disruption of these structures has the potential to alter downstream gene regulation, as shown by numerous studies. However, compartments have been found, more so than TADs, to be strongly associated with cell type and cell fate. Therefore, in this manuscript, we decided to focus only on the compartment organization changes between different healthy and cancerous cells as they are more likely to represent the stable alterations of the genome organization malignant transformations.

(7) In the final sentence of the discussion the authors stated "Overall, our results suggest that genome spatial compartment changes can help encode a cell state that favors metastasis (EMT)". The "metastasis (EMT)" was in fact not clearly linked inside the manuscript. The authors did not provide a strong link between metastasis and EMT in their result description. It is also unclear whether the EMTassociated compartment identity would also correlate with the organotropic compartment identity.

We agree that this statement involves too strong of an assumption. The literature on this topic is vast and complex, and while there is abundant evidence that pathways of EMT can play important roles in facilitating metastasis, there are other pathways at play in the metastatic process as well (https://journals.plos.org/Plosbiology/article?id=10.1371/journal.pbio.3002487). We have made a clearer statement about this in the text now.

To address the question of whether the organotropic changes related to the EMT changes, we calculated the overlap between the genomic bins that strongly segregated cell lines in the compartment principal component analysis (PC1) with those that showed “organotropic” changes. As you can see in supplementary table 3, this overlap is actually very small, where only 3% of bins are important both for the EMT segregation of cell lines and organotropism.

We have now included this overlap information as supplementary table 3 and have addressed this in the text.

**Reviewer #2 (Public review):**
Summary:This work addresses an important question of chromosome architecture changes associated with organotopic metastatic traits, showing important trends in genome reorganization. The most important observation is that 3D genome changes consistent with adaptations for new microenvironments, including lung metastatic breast cells exhibiting signatures of the genome architecture typical to a lung cell-like conformation and brain metastatic prostate cancer cells showing compartment shifts toward a brain-like state.Strengths:This work presents interesting original results, which will be important for future studies and biomedical implications of epigenetic regulation in norm and pathology.Weaknesses:The authors used publicly available data for 15 cell types. They should show how many different sources the data were obtained from and demonstrate that obtained results are consistent if the data from different sources were used.

In our revised version, we have provided a clarified table of information about all the publicly available data used from all the cell lines, indicating the sources of the data. The 17 datasets used come from 8 different studies. So, indeed, the reviewer is correct that many different sources of data were used. To address the question of whether our results would be consistent if data from different sources were used, we created a comparison map of the A/B compartment profiles for data from multiple sources when it was available. You can see below that the Hi-C data from different sources for the same cell lines cluster quite closely and show high correlation and are well separated from different cell lines. So, we do not think that source batch effects play a major role in our results.

**Author response image 3. sa3fig3:** 

**Recommendations for the authors:**

**Reviewer #1 (Recommendations for the authors):**
(1) Figure 1a: This figure could be re-formatted without the arrows. Arrows usually indicate upstreamto-downstream relationships along certain processes. Using arrows here would mislead people to think that the cell lines were derived from one another. The same could apply to the supplementary figures.

We have now edited figure 1a to include lines linking cell lines, indicating conceptual relationships, rather than arrows, which would imply direct derivation.

(2) Figure 1c: The PCA (PC2 axis) indeed seemed to separate the HER2 status quite well. One concern is MCF7, it is labeled as ERpos/HER2neg in MetMap but seems to be clustered as HER2pos in this study. Are they the same? (This again highlights the importance of cell line definition and annotation).

It is a good point that MCF7, while generally considered HER2 negative (we indicate this negative status in Supplementary Table 1), falls near HER2 positive cells in PCA space. This indicates that PCA captures tendencies but is not a perfect classifier. In a high dimensional, complex system, it is expected that an unsupervised analysis such as this will not capture just one biological feature in a given principal component, and therefore something like HER2 status may not segregate perfectly. However, this analysis does suggest that MCF7 3D genome structure has features that are more similar to other HER2+ cell lines. This raises the interesting possibility that it may actually behave like HER2+ cells in some ways even while being HER2- itself. We have more clearly stated the MCF7 discrepancy in the text.

**Reviewer #2 (Recommendations for the authors):**
(1) The description of results can be shortened, to make it easier to read and understand.

In our revision, we have tried to clarify where possible, but it was difficult to shorten without losing important caveats and context (especially to make important points emphasized by reviewer 1).

(2) "100 most positive and negative eigenvalues for PC1" - please provide the correct description.

We have altered this to make it clearer and more correct: “using the genes from the regions with the top 100 most positive and 100 most negative eigenvector loadings for this PC1”